# Digital Image Correlation (DIC) Assessment of the Non-Linear Response of the Anterior Longitudinal Ligament of the Spine during Flexion and Extension

**DOI:** 10.3390/ma13020384

**Published:** 2020-01-14

**Authors:** Maria Luisa Ruspi, Marco Palanca, Luca Cristofolini, Christian Liebsch, Tomaso Villa, Marco Brayda-Bruno, Fabio Galbusera, Hans-Joachim Wilke, Luigi La Barbera

**Affiliations:** 1Department of Industrial Engineering, School of Engineering and Architecture, Alma Mater Studiorum—Università di Bologna, I-40131 Bologna, Italy; marialuisa.ruspi2@unibo.it (M.L.R.); marco.palanca@unibo.it (M.P.); 2Institute of Orthopaedic Research and Biomechanics, Trauma Research Center Ulm (ZTF), University Hospital Ulm, D-89081 Ulm, Germany; christian.liebsch@uni-ulm.de (C.L.); hans-joachim.wilke@uni-ulm.de (H.-J.W.); 3Laboratory of Biological Structure Mechanics, Department of Chemistry, Materials and Chemical Engineering “G. Natta”, Politecnico di Milano, I-20133 Milan, Italy; tomaso.villa@polimi.it (T.V.); luigi.labarbera@polimi.it (L.L.B.); 4IRCCS Istituto Ortopedico Galeazzi, I-20161 Milan, Italy; fabio.galbusera@grupposandonato.it; 5Department of Spine Surgery III, IRCCS Istituto Ortopedico Galeazzi, I-20161 Milan, Italy; marco.brayda@gmail.com; 6Department of Mechanical Engineering, Polytechnique Montréal, Montréal, QC H3T 1J4, Canada; 7Research Center Sainte-Justine University Hospital Center, Montréal, QC H3T 1C5, Canada

**Keywords:** spine biomechanics, anterior longitudinal ligament (ALL), digital image correlation (DIC), in vitro testing, non-linear material properties

## Abstract

While the non-linear behavior of spine segments has been extensively investigated in the past, the behavior of the Anterior Longitudinal Ligament (ALL) and its contribution during flexion and extension has never been studied considering the spine as a whole. The aims of the present study were to exploit Digital Image Correlation (DIC) to: (I) characterize the strain distribution on the ALL during flexion-extension, (II) compare the strain on specific regions of interest (ROI) of the ALL in front of the vertebra and of the intervertebral disc, (III) analyze the non-linear relationship between the surface strain and the imposed rotation and the resultant moment. Three specimens consisting of 6 functional spinal units (FSUs) were tested in flexion-extension. The full-field strain maps were measured on the surface of the ALL, and the most strained areas were investigated in detail. The DIC-measured strains showed different values of peak strain in correspondence with the vertebra and the disc but the average over the ROIs was of the same order of magnitude. The strain-moment curves showed a non-linear response like the moment-angle curves: in flexion the slope of the strain-moment curve was greater than in extension and with a more abrupt change of slope. To the authors’ knowledge, this is the first study addressing, by means of a full-field strain measurement, the non-linear contribution of the ALL to spine biomechanics. This study was limited to only three specimens; hence the results must be taken with caution. This information could be used in the future to build more realistic numerical models of the spine.

## 1. Introduction

The Anterior Longitudinal Ligament (ALL) represents a band covering the anterior aspect of the spine from cervical to lumbo-sacral levels [1]. It contains a high proportion of stiff closely packed collagen fibers, in contrast with other spinal ligaments with more elastin [2]. The most superficial longitudinal fibers span multiple functional spinal units (FSUs), while the deepest are integrated on the periosteum of adjacent vertebrae and cannot be histologically distinguished from the annulus fibrosus of the intervertebral disc (IVD) [3,4]. The collagen fibers, initially broadly organized with a wavy pattern in the unloaded state, are progressively recruited, aligned and stretched along the loading direction, thus yielding a characteristic highly non-linear mechanical response [2].

Experimental studies at the tissue level on isolated ALL and on vertebra–ALL–vertebra specimens reported a highly non-uniform and non-linear biomechanical response. In addition, the failure stress decreases with age and disc degeneration [5,6]. The stiffness and the failure tensile force increased with bone mineral content [7,8]. Viscoelasticity of isolated ALL has also been reported [1,2,6,8]. Several authors have also demonstrated the presence of longitudinal and transversal pre-strains on the ALL in vivo [1,2,9].

In vitro studies on single FSUs have demonstrated that the ALL is almost linearly loaded in flexion-extension and contributes to stabilize the FSU, while protecting the spinal cord from excessive strain [10,11]. Step-wise reduction studies on single thoracic and lumbar FSUs, highlighted the fundamental stabilizing effect of the ALL in extension both following a posterior-to-anterior [12] and an anterior-to-posterior resection protocol [13]. Moreover, the ALL contributes to stabilizing the IVD and constraining the anulus fibrosus from excessive bulging, primarily during flexion [12].

Although previous studies provide some insights on the biomechanical role of the ALL, they present some intrinsic limitations. Analyses at tissue level require the disruption and/or the dissection of the ligamentous structures, therefore neglecting the potential interplay with the surrounding structures (vertebrae and IVDs) on the local mechanical response. When a single FSU is tested, only a short portion of the ALL is included: this preserves the deep fibers (connecting two adjacent vertebrae) but not the superficial ones (spanning over several vertebrae). Therefore, only the contribution of the short fibers is properly assessed, while the effect of the long ones is partly compromised. In fact, in vitro tests on long multi-segmental spinal segments pointed out the importance of preserving the integrity of the ligamentous structures extending over multiple FSUs to correctly catch the complex behavior of the spine [14].

Although a topographical description of the strain values along the length and the width of the ALL has already been reported on multiple lumbar FSUs, the evaluation was based on only a small number of discrete points [10]. Another analysis based on a laser scanner, performing 3-d surface scans, allowed an indirect evaluation of the local strains of the IVD under complex loading [15], but it completely neglected the ALL. More recently, Digital Image Correlation (DIC) overcame these limitations, allowing the evaluation of the full-field strain distribution on the entire surface of the ALL under different loading conditions [16,17,18].

The present in vitro study addressed the behavior of the superficial layers of the ALL using full-field DIC analysis on intact multi-segmental spinal specimens. The focus was on the lumbar spine, specifically on the L4-L5 region, where the greatest number of soft tissue lesions are reported (such as disc herniation) [19]. This is a basic science study aiming to provide data about the non-linear contribution of the ALL during the different phases of spinal flexion and extension. This information could provide identification more detailed criteria to build better multibody spine models able to capture the changing stiffness of the ALL during motion, or to include more realistic material properties in finite element models. The specific aims were to:
Characterize the strain distribution of the ALL in situ during flexion-extension;Compare the strain in specific regions of interest (ROIs), in front of L4 vertebral body (VB) and L4-L5 IVD (this region of the spine is subject to greater range of motion);Analyze the non-linear relationship between the measured strain and the imposed rotation and the resultant moment.

## 2. Materials and Methods

### 2.1. Specimens

To analyze the behavior of the ALL, three fresh-frozen human thoracolumbar spine segments (consisting of 6 FSUs from T12 to sacrum) were obtained through an ethically approved international donation program (Science Care Inc., Phoenix, AZ, USA) (Figure 1). The donors were all Caucasian, two males and one female (Table 1). Clinical computed tomography (CT) scans (Philips Brilliance 64, Philips Healthcare, Cleveland, OH, USA, with a resolution of 0.4 mm) were used to verify the state of degeneration and to determine the bone mineral density (BMD) [20]. No fractures, tumors were observed; however, all specimens showed some osteophytes, as can be expected with aged donors [21].

The spines were carefully cleaned on the anterior side, removing fat tissue and muscles in order to expose the ALL, while all the posterior osteo-ligamentous structures were left intact. The two extremities of the specimens were potted in poly methyl-methacrylate cement (PMMA, Technovit 3040, Heraeus Kulzer, Werheim, Germany). Specimen hydration was granted by spraying saline solution on their surface during the tests.

### 2.2. Mechanical Test

The load was applied using a state-of-the-art spine tester [22,23]: the caudal side was fixed, while the cranial side was connected to the gimbal with three integrated stepper motors (FT 1500/40, Schunk GmbH & Co. KG, Lauffen am Neckar, Germany) (Figure 1). A six-component load cell measured the moments and forces applied. All specimens were tested in flexion-extension (up to ±7.5 Nm at a rate of 1°/s) at room temperature (ca. 23 °C) [23]. All motions started and finished in the unloaded neutral position [23]. Each test consisted of three consecutive cycles of loading: the first two cycles for pre-conditioning and the last one for the actual analysis [23]. 

### 2.3. Measurements of Intervertebral Motions

The different levels of the spine exhibit different values of flexibility, range of motion (RoM) and stiffness. The analysis focused on lumbar spine (in detail on L4–L5 vertebrae) because this region of the spine is more subject to pain especially due to soft tissue lesions (such as disc herniation) [19]. Therefore, in order to measure the RoM in terms of angle between L4 and L5, an optical motion tracking system was used. Six cameras (MX13, Vicon Motion Systems Ltd., Oxford, UK) quantified the 3D coordinates of three reflective markers positioned on L4 and L5 vertebrae. Starting from these data, the L4–L5 intervertebral angle was calculated in the sagittal plane during the three loading cycles.

### 2.4. Digital Image Correlation

The test (three additional loading cycles) was repeated after removing the markers, using DIC to obtain a full-field strain distribution on the ALL. A white-on-black speckle pattern was prepared on the anterior surface of the specimens. The multi-segmental spine segments were first stained with a 4% solution of methylene blue and water [16,24,25]. The white speckle pattern was applied using an airbrush gun following an optimized procedure [26]. This method has been demonstrated to not significantly affect the biomechanical behavior [17,26].

A commercial 3D-DIC system was used (Q400, Dantec Dynamics, Skovlunde Denmark) with its software (Instra 4D, v. 4.3.1, Dantec Dynamics, Skovlunde, Denmark), equipped with two cameras (5 MegaPixels, 2440x2050, 8-bit, black-and-white) with 17 mm lenses. The specimens were illuminated with a system of LEDs (10000 lumens in total). The field of view was set to 120 mm by 160 mm which gave a pixel size of about 0.08 mm and a depth of field of 70 mm with the adopted aperture (f/22). Images were acquired at 5 frames per second. Calibration was performed before the tests using a proprietary calibration target (Al4-BMB-9x9, Dantec Dynamics, Skovlunde, Denmark).

The main source of error in DIC-measured strain derives from the image noise, which translates to random strain errors [24,27]. Therefore, the parameters for the correlation analysis were preliminarily optimized during a zero-strain test for each specimen to minimize the errors (Table 2).

### 2.5. Analysis of Strain

The distribution of strain was evaluated on the ALL in front of the L4 vertebra and in front of the L4–L5 IVD. To focus on the ALL and investigate its mechanical contribution, regions of interest (ROIs) were identified in correspondence with the most strained areas of the ALL. The spots with the peak longitudinal strain (ε_long_) were first identified, both in front of the vertebra and in front of the IVD. The two ROIs were then selected on each specimen so as to include the area around such spots where strains were higher than 50% of the corresponding peak previously identified. In this way, roughly rectangular areas of about 200–250 mm^2^ were identified. For each ROI, the values of longitudinal strain were analyzed throughout the entire load cycles as the mean over the ROIs.

### 2.6. Analysis of the Non-Linearity

To analyze the non-linear behavior of the spine segment, first of all the neutral zone (NZ) and elastic zone (EZ) were identified on the third cycle of the moment-angle curve. The slope of the curve was calculated throughout the load cycle with a moving linear regression on 10 points (which corresponded to 2%–4% of the total points of the curve, and to an interval of 2–3 s).

The limit of the NZ was defined to be where the slope became greater than 2 Nm/deg. The EZ limit was defined from the end of NZ and to the peak of the moment-angle curve. The NZ and the EZ zone were identified for both directions of motion (from flexion to extension and from extension to flexion).

### 2.7. Assessment of Measurement Uncertainties

The measurement uncertainties were evaluated with preliminary analysis:
Range of motion (from the Vicon system): the error on the measurement of the angle between L4–L5 was less than 0.1° [29].Strain uncertainty (DIC system): as the largest component of error in DIC-measured strain is the random error [24,27], a zero-strain analysis was used. Two images of each unloaded specimen were captured with the DIC system and analyzed with the optimal software parameters to evaluate the strain measurement uncertainties in a known configuration (zero-strain) [24]. Being in a zero-strain configuration, any strain different from zero was accounted as measurement error. DIC-measured strains had a systematic error less than 0.002% and a random error less than 0.006%.Intra-operator variability: in order to analyze the reliability of ROIs identification, the same operator was asked to identify the ROIs three times on different days in correspondence. The difference among the three repetitions was less than 0.2% of the mean value inside the ROI.

These values were considered satisfactory if compared to the typical rotations (of the order of 10° from full flexion to full extension), and to the typical strain peaks measured in the ALL (4%–6%).

## 3. Results

### 3.1. Range of Motion and Strain Maps

For all the specimens, there was an asymmetry in the RoM between flexion and extension when the same moment of ±7.5 Nm was applied (Figure 2):
During flexion, the L4–L5 angle reached 4.0° for specimens A and B, and 6.3° for specimen C;During extension the angle reached 1.5° for specimen A, 1.7° for specimen B and 2.7° for specimen C.

The full-field strain maps were computed using DIC for all the specimens throughout the tests. During extension, the longitudinal strain was positive (traction) while during flexion it was negative (compression) (Figure 3). The opposite behavior was found for the circumferential strain: in extension the values were negative, showing a circumferential narrowing of the ligament, while in flexion the values were positive showing a transversal stretching due to IVD bulging [18].

The strain distributions on the ALL in correspondence with L4 vertebra and L4–L5 IVD were significantly different: in general on the vertebra the largest strain was in the order of ±1.5%; on the IVD the largest strain was in the order of ±4%. However, for all the specimens, comparable strain was measured on average in the ROIs in front of L4 and in front of the IVD (the longitudinal strain averaged 1.5% for extension and −1.5%: −1% for flexion, while the circumferential strain averaged 3% for flexion and 2% for extension).

While a similar trend was observed in most cases, a different behavior was seen for specimen A with respect to the strain in front of the IVD during flexion (in this ROI the strains were one order of magnitude lower than for the other specimens). Also, in the full-field maps, specimen A showed positive strains in correspondence with the central part of the IVD.

### 3.2. Non-Linear Trend of the Strain in the Different Parts of the ALL

The slope of the final part of EZ was similar in flexion and in extension (Figure 2). The slope of the EZ was one order of magnitude larger than the slope in NZ for all three specimens. 

The limits of the EZ and NZ were reported on strain-moment and angle-strain curves so as to match the points previously identified in the moment-angle curves (Figure 4 and Figure 5). The strain-moment curves (Figure 4) showed a non-linearity similar to the moment-angle curves (Figure 2). In fact, the strains in the EZ grew (in terms of slope of the strain-moment plots) 1–2 orders of magnitude slower than in the NZ. Furthermore, the non-linear trend of the strain-moment curves was not symmetrical: in flexion the change of slope of the curve was much more abrupt than in extension. In addition, the difference between the slope of the NZ and EZ was much more pronounced in front of the vertebra (roughly a factor 4) than in front of the disc (factor 2). Conversely the strain-angle curves showed a more linear trend, with much smaller changes of slope (Figure 5). In fact, the ratio between the slope of the NZ and the slope of the EZ was between 0.3 and 3.

In Table 3, the values of moment, angle and strain on vertebra and IVD are reported separately for the flexion-to-extension and extension-to-flexion curves in correspondence of the points which identified the transition from NZ to EZ. These two regions were not symmetric respect to the moment: during flexion the EZ started at a moment between 2.6 and 4.0 Nm while during extension the EZ started at a moment value around 0 Nm. For what concerns the angles, in flexion the EZ started at 2.6°–5.6° while in extension it started at 0.2°–2.0°.

To verify whether there was a correlation between a variation in the longitudinal strain on the ALL in front of the vertebra and in front of the L4–L5 IVD, these variables were plotted in the same graph (Figure 6). This trend was close to a straight line, with a slope close to 1.0 (in the range of 0.8–1.2). 

All specimens had the same behavior, again with the exception of specimen A during flexion, which showed no change in the strain values on the IVD when the strain on the VB reached −2%. Conversely during extension, specimen A showed the same behavior as the other specimens.

## 4. Discussion

The aim of this study was to investigate the behavior of the ALL using full-field DIC analysis on intact multi-segmental spinal specimens, so as to gather data about the non-linear contribution of the ALL during the different phases of spinal flexion and extension. This information is currently missing in the literature, and could contribute to building better multibody models, and better finite element models of the spine, able to capture the changing stiffness of the ALL during motion. 

A flexion-extension test was performed measuring the range of motion, the neutral zone, and the elastic zone of the spine, and the strain distribution over the superficial fibers of the ALL. The use of long segments of spine (6 FSUs) allowed to preserve the continuity of the ALL whose superficial fibers span over several vertebrae and IVDs [1,2,9]. A full-field strain distribution on the ALL in front of L4 vertebra and in front of L4–L5 IVD was measured using an established DIC tool [17,18,24]. In the present study, in order to analyze the non-linear behavior of the ALL, specific regions of interest (ROIs) were identified on its surface in front of the vertebra and in front of the IVD, and for each region the longitudinal strains were analyzed.

In general, ligament fibers transmit only tensile forces. The ALL stretches longitudinally in extension, but works in traction too (in circumferential direction) when the spine is in flexion. During this movement, the IVD is compressed and bulges, transmitting tension to the ALL [4]. In fact, the role of the ALL is to limit the movement of the spine during extension [11,30]. Similarly, the Posterior Longitudinal Ligament (PLL) limits the movement of the spine in flexion [4,13]. In addition, longitudinal ligaments (ALL and PLL) are much stiffer than the other ligaments (such as ligamentum flavum): longitudinal ligaments are closer to the neutral bending axes and so in order to provide the same moment, the stiffness must be greater [2,31,32]. 

In this work, during extension, the analysis of strain maps (both in correspondence with vertebra and IVD) highlighted a stretch of the superficial layers of the ALL in the longitudinal direction. This is in accordance with literature, reporting that the fibers of the ALL are aligned longitudinally with the axes of the spine [2]. Conversely, during flexion, the strain maps showed negative longitudinal strains, which highlighted a shortening of the ALL in the same direction. At the same time, during flexion there was a stretching in the circumferential direction [18].

To deeply investigate the local spinal RoM, the moment-angle curves were calculated using the testing machine and Vicon system. These curves were asymmetrical, and the spine reached higher intervertebral rotations during flexion than during extension (6° degrees respect to 2.7° degrees for the specimen with the highest RoM). These findings are compatible with previous studies on the spinal range of motion [23,33,34]. Furthermore, the moment-angle curves showed a very accentuated non-linear behavior as already described in the literature [34]. The slope of these curves changed more abruptly in flexion, while on the contrary the slope changed more damped in extension. The EZ zone was larger for extension than for flexion, in fact the beginning of EZ during flexion corresponded to values of moment and angle greater than during extension. As soon as the column moves towards extension, the ALL is stretched and limits this movement immediately but with a gradual slope (in fact the slope of the curve is smaller in extension respect to flexion). This may be due to the mechanical role of the facet joints and the capsular ligaments, which contribute with the ALL to stabilize extension [35]. Conversely, during flexion the lumbar spine allowed more degrees of movement in flexion before stiffening. This may be discussed in consideration of the fact that higher rotations may be needed to achieve the high tensile longitudinal pre-stretch typical of the ALL [1,2,10] before reaching negative values during bulging of the annulus [12].

Like for moment-angle curves, the strain on ROIs in front of the vertebra and in front of the IVD also showed a non-linear and asymmetric trend for flexion and extension. This behavior is in agreement with the non-linear behavior measured in isolated spinal ligaments in the past [5,31]. The ALL consists of fibers which are pre-stretched both in longitudinal and transverse direction over the column and resist immediately in extension [2,11]. Conversely, there are other ligaments that are slack and limit the movement only when certain angles are reached [2,11]. At the beginning of the movement, in the NZ, the first fibers that are recruited are the elastin fibers which however do not contribute in giving great resistance. For this reason, large variations of strain were measured in the NZ with small variation of moment. Subsequently, collagen fibers (which are stiffer) were also recruited contributing in this way in limiting the movement.

Specimen A showed a different behavior compared to the other specimens for what concerns the trend of strain only during flexion. The strain maps showed that the IVD did not bulge during flexion and so in this way the ALL was not stretched in circumferential direction avoiding the decrease of its length longitudinally; this differs from the behavior usually observed in healthy spines [30]. This was associated with the presence of osteophytes [20] on the endplates of adjacent vertebrae [18], which may alter the load transfer on the ALL. Although specimens B and C also demonstrated some degenerative signs, the presence of lower grade osteophytes was noticed only on one endplate. The correlation between the longitudinal strain values on the ROIs in front of the vertebra and the IVD demonstrated a linear relationship with a slope of about 1.0. This meant that, in the most deformed areas of the superficial layers, the ligament was deformed in the same way both on vertebra and on IVD. This could be further proof of the effect of the longitudinal arrangement of the ALL, which influenced its deformation both in front of the vertebra and in front of the IVD. It is true that the deeper layers are attached to the vertebra and to the external layers of the IVD; conversely, the most superficial layers of the ALL are not constrained in such a way, and are affected by the effect of the adjacent joints and the longitudinal extension of the fibers of this ligament, as documented in the past [2,3,8].

A limitation of the present work is the small number of specimens tested (N = 3), which was constrained by the extensive measurement campaign required by this DIC analysis. For this reason, the present findings must be taken with caution. For instance, specimens from other groups of donors (e.g., younger, or with severe deformity) could yield different results. However, the behavior of the ALL was similar for all specimens. Furthermore, only direction of loading (flexion and extension) was considered in this study. A pure moment was applied so as to deliver a highly controlled loading. The rationale is that the ALL constraints mainly the movements on the sagittal plane limiting the extension, while it gives a marginal contribution during lateral bending or axial torsion. A further limitation relates to the relatively slow motion imposed (1°/s). This condition was chosen for consistency with previous similar studies [30].

This could be a starting point for other studies in which the behavior of other ligaments could be investigated under different loading conditions or after surgical interventions on the spine [36].

## 5. Conclusions 

DIC analysis has been shown to be a valid tool for measuring the strain in a full-field way without altering the complex structure of the spine. In conclusion, to study the behavior of the ALL, it is important not to separate it from the other structures but to consider the column as a whole. In addition, the non-linear response of the ALL between the neutral zone and the elastic region, and its different behavior in flexion and extension should be considered for instance for multibody modeling of the spine kinematics of finite element models investigations.

## Figures and Tables

**Figure 1 materials-13-00384-f001:**
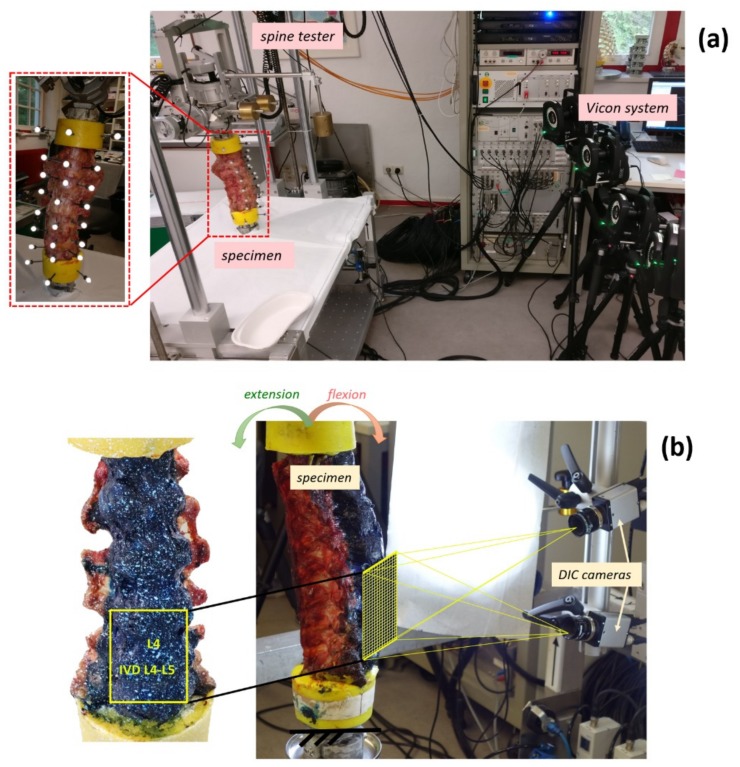
Overview of the test configuration and data acquisition systems. The top part (**a**) shows the test session where motion was measured: the spine segment with the markers (three on each vertebra) is visible on the left. The spine tester and four of the six cameras of Vicon system are visible on the right. The bottom part (**b**) shows the session where strains were measured: the specimen with the white-on-black speckle pattern for the DIC analysis is visible on the left. The field of view recorded by the DIC cameras is indicated. On the right, the specimen mounted in the spine tester in front of the DIC system is shown.

**Figure 2 materials-13-00384-f002:**
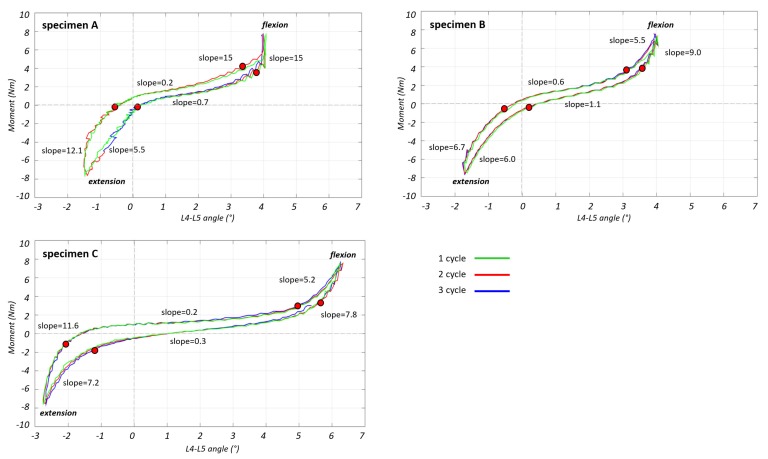
The moment-angle curves for the three loading cycles are reported for specimens A, B and C. Positive values of moment correspond to flexion, while negative values of moment correspond to extension. The circles indicate the end of the NZ and the beginning of the EZ (identified where the slope of the curve reached 2 Nm/deg). The values of minimum and maximum slope for the NZ and EZ are reported for both of the curves flexion to extension and extension to flexion.

**Figure 3 materials-13-00384-f003:**
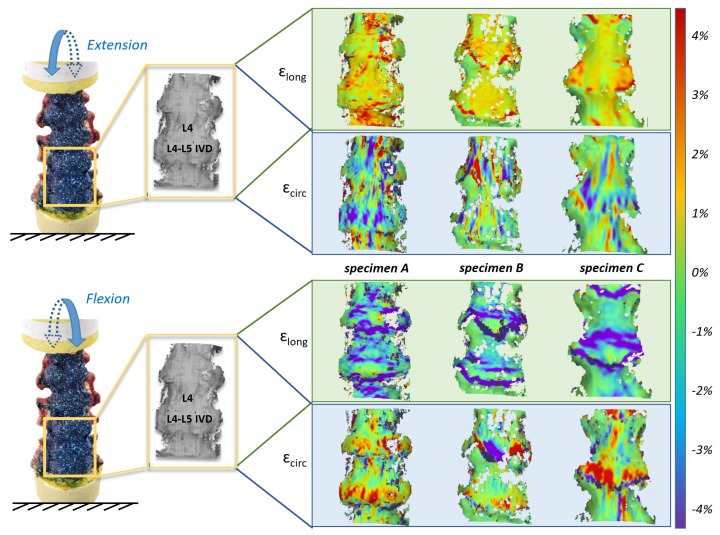
Full-field strain maps obtained from the DIC analysis for all the specimens at the peak load (±7.5 Nm). The longitudinal (ε_long_) and circumferential (ε_circ_) percent strain are reported for extension (top) and flexion (bottom). The color maps show the non-uniform distribution of strain.

**Figure 4 materials-13-00384-f004:**
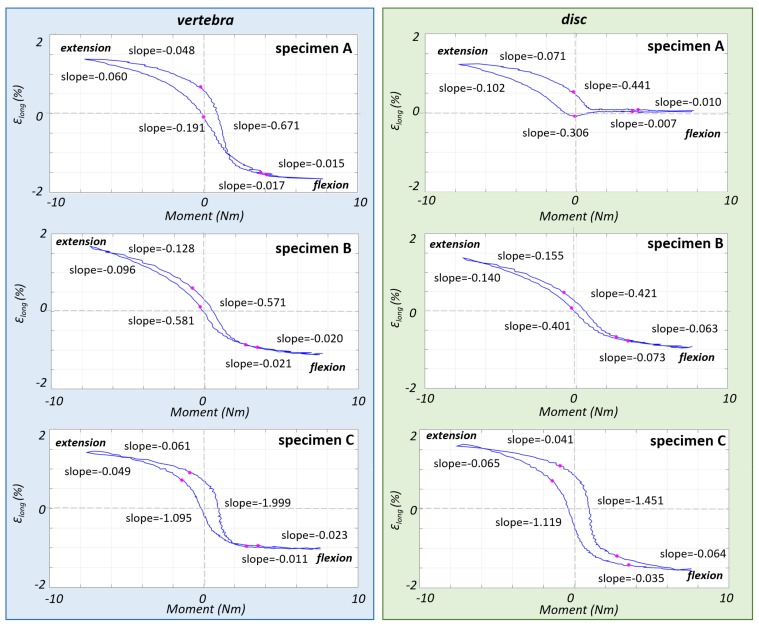
The strain-moment curves of the third loading cycle are reported for each specimen, showing how the strain in the ALL varied in front of the vertebra (left charts) and in front of the disc (right charts) for specimens A, B, C (from top to bottom). The end of the NZ and the beginning of the EZ (identified on moment-angle curves, Figure 2), are reported here with circles. The values of minimum and maximum slope (% strain/Nm) for the NZ and EZ are reported for both of the curve flexion-to-extension and extension-to-flexion.

**Figure 5 materials-13-00384-f005:**
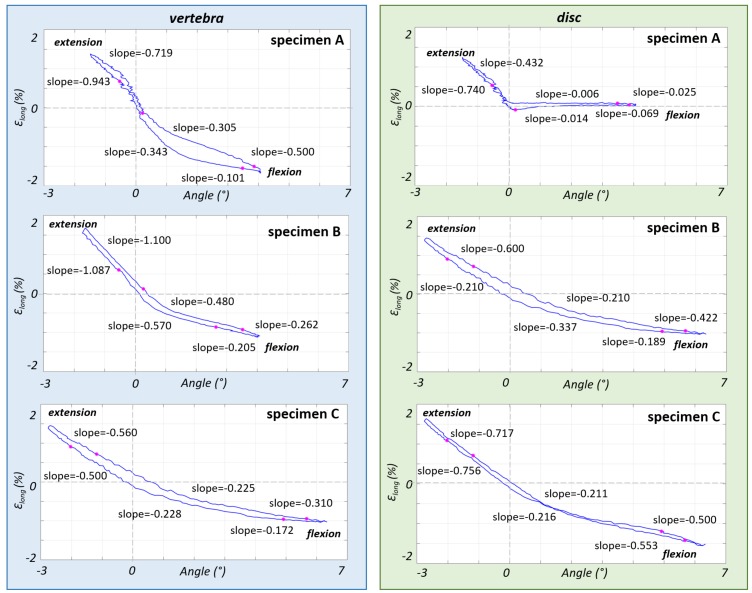
The strain-angle curves of the third loading cycle are reported for each specimen, showing how the strain in the ALL varied in front of the vertebra (left charts) and in front of the disc (right charts) for specimens A, B, C (from top to bottom). The end of the NZ and the beginning of the EZ (identified on moment-angle curves, Figure 2), are reported here with circles. The values of minimum and maximum slope (% strain/° degree) for the NZ and EZ are reported for both of the curves flexion-to-extension and extension-to-flexion.

**Figure 6 materials-13-00384-f006:**
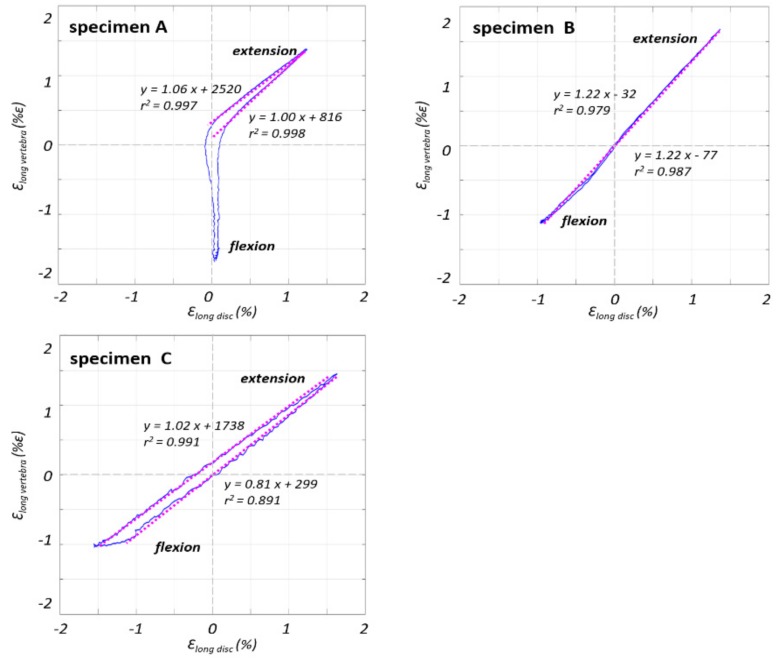
The strain in the ALL in front of the vertebra is plotted against the strain in front of the disc for the third loading cycle, showing a highly linear trend. The regression line is reported with its equation and the r square, separately, for the flexion-to-extension and extension-to-flexion curves.

**Table 1 materials-13-00384-t001:** Details of the specimens with the donor’s information. The last column reports the grading of the osteophytes (scored according to [20]).

Specimen	Segment	Sex	Age at Death (Years)	Height (cm)	Weight (kg)	BMI (kg/m^2^)	BMD (mg/cm^3^)	Assessment of the Osteophytes in the L4–L5 Area
A	T11-S1	M	66	183	141	42.1	82	2 osteophytes (both grade 2) centrally located on the endplate of both L4 and L5
B	T11-S1	M	62	178	164	51.7	94	1 osteophyte (grade 2) centrally located on the endplate of L5
C	T12-S1	F	63	157	125	50.7	157	2 osteophytes (grade 1 and grade 2) centrally located on the endplate of respectively L4 and L5

**Table 2 materials-13-00384-t002:** Details of the parameters used for the correlation analysis with the DIC system (according to [28]).

Parameters for the Correlation Analysis	
DIC Software Package Name and Manufacturer	Instra 4D, v. 4.3.1, Dantec Dynamics
Distance of the cameras	540 mm
Field of view	about 120 mm by 160 mm
Depth of field	70 mm
Lens aperture	f/22
Frame rate	5 frames per second
Grid spacing	4 pixels
Facet size	between 39 and 59 pixels
Pixel size	about 0.08 mm
Contour smoothing	kernel size 5 × 5

**Table 3 materials-13-00384-t003:** The values of moment, angle between L4 and L5 and strain (in front of the vertebra and in front of the disc) in correspondence to the end of LZ and the beginning of EZ are reported in the range of angles corresponding to spine flexion and extension, both for the flexion-to-extension (f–e) direction and for the extension-to-flexion (e–f) direction.

Specimen	Moment (Nm)				Angle (°)				Strain in Front of Vertebra (%)				Strain in Front of Disc (%)			
	Flexion		Extension		Flexion		Extension		Flexion		Extension		Flexion		Extension	
	f–e	e–f	f–e	e–f	f–e	e–f	f–e	e–f	f–e	e–f	f–e	e–f	f–e	e–f	f–e	e–f
A	3.70	4.09	−0.10	−0.20	3.86	3.48	0.21	−0.53	−1.55%	−1.50%	−0.14%	0.73%	0.04%	0.08%	−0.08%	0.54%
B	3.42	2.65	−0.31	−0.82	3.52	2.65	0.27	−0.53	−0.92%	−0.86%	−0.12%	0.66%	−0.77%	−0.67%	0.08%	0.48%
C	3.52	2.77	−1.42	−0.92	5.68	4.92	−1.18	−2.03	−0.95%	−0.96%	0.97%	0.79%	−1.42%	−1.19%	1.09%	0.71%

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
