# Peer review of "Digital Image Correlation (DIC) Assessment of the Non-Linear Response of the Anterior Longitudinal Ligament of the Spine during Flexion and Extension"

_materials, 2020, doi:10.3390/ma13020384_

Round 1

Reviewer 1 Report

In this manuscript, Ruspi et al. developed a DIC-based technique to determine the behavior of the Anterior Longitudinal Ligament (ALL) and its contribution during flexion 24 and extension. The method appears to be useful for the full-field strain measurement of the spine. However, this study uses only three specimens and hence its applicability as a general methodology is questionable. The authors may extend the method to several other specimens and compare results with those obtained from other techniques.

Reviewer 2 Report

In this work, a Digital Image Correlation (DIC) system was utilized to study the mechanical behavior of Anterior Longitudinal Ligament (ALL). Specifically, the strain distribution over the whole ALL and also on specific regions of interest during flexion-extension was characterized. The non-linear relationship of applied strain and moment, angle and moment were discussed. At last, the author concluded that the ligament was deformed in the same way on both vertebra and IVD. Background introduction, experimental design and result discussions of this work are complete. I would suggest publication if the author could address my questions below:

Is there any specific reason that L4 and L4-L5 IVD regions are selected?  For section 2.7 (Page 6 of 14) strain uncertainty justification: Other than zero-strain, I think more "standard points" may be necessary here. Can the author comment on that? 

Reviewer 3 Report

General impression

In this article, the authors investigated the biomechanical characteristics of the anterior longitudinal ligament (ALL) utilizing Digital Image Correlation (DIC). And they concluded that this is the first study addressing the non-linear contribution of the ALL to spine biomechanics by means of a full-field strain measurement. As they mentioned, I also believe the information in this study must be useful for the researcher in the spine biomechanics.

The methodology of this study was precisely explained and acceptable. Also, the limitations of this research were well indicated. For these reasons, I think this manuscript is appropriate for publication.

However, I have a couple of minor requests to be revised as stated below. After they have been well resolved, I will judge this manuscript can be accepted and published by the materials journal.

abbreviation of DIC: title

I could not easily understand the total words of “DIC”. I recommend the total words of DIC will be indicated in the title because some readers may not understand it, too.

“Digital Image Correlation (DIC) assessment of the non-linear…..flexion and extension”

Abstract line 38

I could not understand of the meaning of “fidelic” and find it in the dictionary. Is it a correct word? It may be replaced by more common word with same meaning.

Discussion line 286

“Digital Image Correlation” here can be abbreviated as “DIC”.

Round 2

Reviewer 1 Report

Recommended acceptance.